# Kinetic Modeling of DUSP Regulation in Herceptin-Resistant HER2-Positive Breast Cancer

**DOI:** 10.3390/genes10080568

**Published:** 2019-07-26

**Authors:** Petronela Buiga, Ari Elson, Lydia Tabernero, Jean-Marc Schwartz

**Affiliations:** 1School of Biological Sciences, Faculty of Biology, Medicine and Health, University of Manchester, Manchester Academic Health Science Centre, Manchester M13 9PT, UK; 2Department of Molecular Genetics, Weizmann Institute of Science, Rehovot 76100, Israel

**Keywords:** kinetic model, breast cancer, Herceptin, dual-specificity phosphatases

## Abstract

Background: HER2 (human epidermal growth factor 2)-positive breast cancer is an aggressive type of breast cancer characterized by the overexpression of the receptor-type protein tyrosine kinase HER2 or amplification of the *HER2* gene. It is commonly treated by the drug trastuzumab (Herceptin), but resistance to its action frequently develops and limits its therapeutic benefit. Dual-specificity phosphatases (DUSPs) were previously highlighted as central regulators of HER2 signaling; therefore, understanding their role is crucial to designing new strategies to improve the efficacy of Herceptin treatment. We investigated whether inhibiting certain DUSPs re-sensitized Herceptin-resistant breast cancer cells to the drug. We built a series of kinetic models incorporating the key players of HER2 signaling pathways and simulating a range of inhibition intensities. The simulation results were compared to live tumor cells in culture, and showed good agreement with the experimental analyses. In particular, we observed that Herceptin-resistant DUSP16-silenced breast cancer cells became more responsive to the drug when treated for 72 h with Herceptin, showing a decrease in resistance, in agreement with the model predictions. Overall, we showed that the kinetic modeling of signaling pathways is able to generate predictions that assist experimental research in the identification of potential targets for cancer treatment.

## 1. Introduction

Breast cancer is a global burden for women’s health that affects one in eight women during their lifetime. The disease typically arises from disruptions to integrated signaling networks in mammary epithelial cells, in which one or more proteins are dis-regulated. The exact mechanisms affected are diverse, and vary among the several breast cancer sub-types. In particular, HER2 (human epidermal growth factor 2)-positive breast cancer is characterized by the overexpression of the receptor-type protein tyrosine kinase HER2 or amplification of the *HER2* gene [1,2,3]. Herceptin (trastuzumab) is a humanized monoclonal antibody that binds to the extracellular domain of HER2 and down-regulates its function, thereby inhibiting tumor cell growth. Herceptin has been suggested to act by promoting HER2 degradation, by inducing antibody-dependent cellular toxicity against the tumor cells that it binds, and by inhibiting downstream signaling pathways such as MAP (mitogen-activated protein) kinases and PI3K (phosphatidylinositol 3 kinase) [4]. Herceptin is widely used for the treatment of HER2-positive breast cancer, but resistance to its action frequently develops and limits its therapeutic benefit [5]. Extracellularly regulated kinases 1 and 2 (ERK1/2), p38, and c-Jun N-terminal kinases 1 and 2 (JNK1/2) act downstream of HER2 [6] and are key players in regulating cell proliferation. Hence, understanding their role in the development of resistance to HER2 is essential. These kinases are activated by the phosphorylation of conserved and neighboring threonine and tyrosine residues, and can be down-regulated by dual-specificity phosphatases (DUSPs), which can dephosphorylate both residues [7]. As such, DUSPs (dual specificity phosphatases) play an important role in regulating these kinases and are themselves potential targets for breast cancer treatment or resistance [8].

Computational modelling has become widely used to help to understand complex signaling processes. By translating biological interactions into mathematical equations, this approach enables the simulation of complex pathways and the prediction of outcomes under a variety of conditions in a time- and cost-efficient manner [9]. Different techniques can be used for computational modelling, which are each best suited to specific types of systems and analysis. For example, Boolean models are well suited to modelling complex signaling pathways when detailed quantitative data are not available [10]. On the other hand, kinetic models provide more precise quantitative information but are more challenging to build and use, given the complexity of the differential equations required. Models of this type can be used to study the effects of varying levels of drugs on cellular systems, to find effects of protein overexpression in cancer signaling, and to evaluate how they can be countered by targeted inhibitions [11].

Given the variability in cellular behavior, kinetic models need to be highly specific to the investigated cancer subtypes and cell types. For example, Qiu et al. (2017) built a kinetic model of regulatory mechanisms between the genes *Zeb1* and *Cdh1*, which are involved in the growth and development of breast cancer cells, providing insights into these mechanisms specifically among the five breast cancer subtypes [12]. Hu et al. (2013) used a kinetic model to show that the overexpression of HER2 can explain the presence of oscillations in mitogen-activated protein kinase (MAPK) and PI3K signaling pathways in breast cancer [13]. Das and Jacob (2018) proposed a kinetic model of the activation of HER receptors by their ligands and downstream signaling pathways and used it to study the efficacy of pertuzumab treatment [14].

Here, we investigated whether inhibiting certain DUSPs re-sensitized Herceptin-resistant breast cancer cells to the drug, using cell numbers as a readout. We achieved this by building a series of kinetic models incorporating the key players of HER2 signaling pathways and simulating a range of inhibition intensities. The results obtained were also examined in live tumor cells in culture. The results reveal varying patterns of effects depending on the DUSP regulation mechanisms.

## 2. Methods

### 2.1. Cell Lines and Transfection

The human breast cancer cell line SK-BR-3 was purchased from the American Type Culture Collection (Manassas, VA, USA). It was grown in McCoy’s 5A medium (Sigma-Aldrich, St. Louis, MO, USA), containing 50 U/mL penicillin, 50 mg/mL streptomycin, and 10% heat-inactivated fetal bovine serum (FBS; Gibco/ThermoFisher Scientific, Waltham, MA, USA) at 37 °C in an atmosphere of 5% CO_2_. 

Human kidney 293T/AD cells were grown in Dulbecco’s Modified Eagle’s Medium (DMEM) (Sigma Aldrich, St. Louis, MO, USA) supplemented with 10% fetal calf serum (FCS) (Gibco), 4 nM glutamine, 50 U/mL penicillin, and 50 µg streptomycin at 37 °C, 5% CO_2_. 

Transient transfection of 293T cells was by BES-calcium phosphate technique [15]. Five hours after transfection, medium was replaced. The cells were allowed to express the desired vectors for 24–48 h and then were collected for RNA extraction and cDNA (complementary DNA) synthesis.

### 2.2. Generation of Herceptin-Resistant Cell Lines

SK-BR-3 cells were cultured in 60-mm plates for two days until they reached 75% confluency, at which time 50 µM Herceptin (Roche, Basel, Switzerland) was added to the medium. Cells were cultured in the presence of Herceptin for six months; during this period the cells were fed every three days with Herceptin-containing medium and split when confluency reached 75%. SK-BR-3 cells cultured in parallel in the same manner but without Herceptin were used as controls. 

### 2.3. Gene Expression Analysis by q-PCR

Total cellular RNA was extracted from cells using the RNeasy Mini Kit (QIAGEN, Hilden, Germany) and treated with DNase I prior to use. One microgram of RNA was reverse transcribed using the qScript cDNA synthesis kit (Quanta Biosciences, Beverly, MA, USA) in a total volume of 20 µL according to the manufacturer’s instructions. A 0.5 µL (25 ng) amount of cDNA was used in quantitative PCR (RT-qPCR) experiments using KAPA SYBR Fast qPCR Master Mix ABI Prism (Kapa Biosystems, Wilmington, MA, USA), together with target-specific primers (as described in [3]). Amplification was performed on an AB StepOnePlus instrument (Applied Biosystems, Foster City, CA, USA). 

Amplification steps started with an activation at 95 °C (20 s), followed by 40 cycles of (95 °C, 3 s; 60 °C, 30 s), and concluding with 1 cycle of (95 °C, 15 s; 60 °C, 60 s; 95 °C, 15 s).

ΔCT (average change in threshold cycle number) values were determined for each DUSP in each sample relative to the housekeeping genes (*β-actin* and *GAPDH*) used as endogenous controls by the ΔΔCT method, in which the fold change in target gene expression is determined relative to a reference sample following normalization to the above two housekeeping genes [16]. DUSP-specific forward and reverse primers were designed using the Primer3 software [17,18], and their efficiencies were assessed by standard curves. Experiments were performed in two biological repeats. Statistical analysis was performed by Student’s *t*-test, using GraphPad Prism v.7.0a for Mac OS X (GraphPad Software, San Diego, CA, USA).

### 2.4. Preparation of VSV-G Lentiviral Particles and Breast Cancer Cells (SK-BR-3) Infection

Lentiviral particles were produced as described [19]. Briefly, 293T cells were seeded on poly-l-lysine and transfected with a commercial vector for expressing the desired shRNA (short hairpin RNA, see below) or with a vector expressing GFP (Green Fluorescent Protein) as control, along with accessory plasmids as stipulated in [19], using the calcium phosphate method [15]. After 5 h, the medium was changed to DMEM-2% FCS (5 mL). Culture medium containing the resulting virions was collected at 24 h and 48 h and was combined. For the infection of cells, sensitive and resistant SK-BR-3 cells were plated in 6-well plates, in 2 mL of complete medium/well. On the following day, the growth medium was replaced with 3 mL/well of medium containing lentiviruses expressing shRNA against specific DUSPs, or non-target (NT) shRNA, along with 8 µg/mL polybrene. After 24 h, viruses were removed, and complete medium was added for another 24 h. The cells then underwent selection in 1 µg/mL puromycin for 3 days, after which puromycin concentration was reduced by 50%. At this point, colonies of resistant cells became evident in cultures infected with virions, but not in uninfected control cells. 

The DUSP shRNAs were purchased from Dharmacon (Lafayette, CO, USA) as glycerol stocks, as follows: DUSP8 shRNA (RHS4533-EG1850) containing five constructs (RHS3979-201909039, RHS3979-201910914, RHS3979-201915816, RHS3979-201908134, and RHS3979-201910445); and DUSP16 shRNA (RHS4533-EG80824) containing five constructs (RHS3979-200805108, RHS3979-200805108, RHS3979-200805109, RHS3979-200805110, and RHS3979-200805111).

### 2.5. Assay for Herceptin Sensitivity

For each repeat, 0.5 × 10^6^ cells were plated in six-well plates and cultured overnight in growth medium lacking Herceptin. Twenty-four hours later, cells present in several wells were counted, establishing the cell count at the start of the experiment. The cells in the remaining wells were then grown for 72 h in the presence or absence of 50 µM Herceptin, after which cells present in these wells were counted, revealing the magnitude of change in cell numbers in the presence of absence of Herceptin. Statistical significance between gene expression and cell counts was established using the Z-test, and was considered significant if *p* < 0.05. We note that the changes in cell numbers following Herceptin treatment that were measured here could arise from changes in cell proliferation and/or cell survival. 

### 2.6. Model Construction

Kinetic models were constructed according to previously described DUSP regulatory interactions [20]. Each protein was represented by two nodes corresponding to its active and inactive form, respectively. The reaction rates were quantified using first-order mass-action kinetics in all reactions. 

Herceptin was assigned a constant value in the model, and its level was set to 100. Other active and inactive protein levels were regarded as variables. Initial amounts of HER2, ERK1 and ERK2 (referred to as ERK1/2), and DUSP8 and DUSP16 (DUSP 8/16) were set to 100; and initial amounts of JNK1 and JNK2 (JNK1/2) and p38 were set to 0, based on previous reports [3]. 

Cell count was represented in the model using a Survival node. The effects of the proteins in the signaling pathways on cell count were quantified using a linear combination of positive and negative components: Activated ERK1/2 (noted ERK12 in the model) is known to favor survival; therefore, a positive mass-action kinetic equation was added from ERK1/2 to Survival. The combination of p38 and JNK1/2 (noted JNK12 in the model) is known to counteract survival; therefore, a negative mass-action kinetic equation was added from associated p38 and JNK1/2 to Survival. The cooperative interaction between p38 and JNK12 was modeled using a modified Hill equation with a coefficient of 0.6. The detailed list of kinetic equations included in the model is shown below:$$\frac{dHER2_{ac}}{dt}=k_{1}\cdot HER2_{i}-d_{1}\cdot HER2_{ac}\cdot Herceptin,$$

$$\frac{dHER2_{i}}{dt}=d_{1}\cdot HER2_{ac}\cdot Herceptin-k_{1}\cdot HER2_{i},$$

$$\frac{dERK12_{ac}}{dt}=k_{2}\cdot ERK12_{i}\cdot HER2_{ac}-d_{2}\cdot ERK12_{ac},$$

$$\frac{dERK12_{i}}{dt}=d_{2}\cdot ERK12_{ac}-k_{2}\cdot ERK12_{i}\cdot HER2_{ac},$$

$$\frac{dJNK12_{ac}}{dt}=k_{3}\cdot JNK12_{i}\cdot HER2_{ac}-d_{3}\cdot JNK12_{ac}\cdot DUSP16_{ac},$$

$$\frac{dJNK12_{i}}{dt}=d_{3}\cdot JNK12_{ac}\cdot DUSP16_{ac}-k_{3}\cdot JNK12_{i}\cdot HER2_{ac},$$

$$\frac{dP38_{ac}}{dt}=k_{4}\cdot P38_{i}\cdot HER2_{ac}-d_{4}\cdot P38_{ac}\cdot DUSP16_{ac},$$

$$\frac{dP38_{i}}{dt}=d_{4}\cdot P38_{ac}\cdot DUSP16_{ac}-k_{4}\cdot P38_{i}\cdot HER2_{ac},$$

$$\frac{dDUSP16_{ac}}{dt}=k_{5}\cdot DUSP16_{i}-d_{5}\cdot DUSP16_{ac},$$

$$\frac{dDUSP16_{I}}{dt}=d_{5}\cdot DUSP16_{ac}-k_{5}\cdot DUSP16_{i},$$

$$\frac{dSurvival}{dt}\;=\;s_{1}\cdot ERK12_{ac}\;-\;s_{2}\cdot {\left(JNK12_{ac}\cdot P38_{ac}\right)}^{0.6}.$$

In the equations, *ac* represents the active form of each protein and *i* the inactive form; *k* represents activation kinetic parameters and *d* inactivation kinetic parameters; *s* represents kinetic parameters affecting the survival function.

Ordinary differential equations (ODEs) were solved using the NumPy package under Python 3. Time-course simulations were run using the odeint function from the SciPy package. DUSP8/16 inhibition was simulated by increasing the *d*_5_ parameter, which represents the rate of transformation from active DUSP8/16 (named DUSP16_ac_ in the model) into inactive DUSP8/16 (named DUSP16_i_ in the model).

## 3. Results

### 3.1. Kinetic Models of DUSP Regulation

Resistance to Herceptin treatment is known to arise in HER2-positive breast tumors.

We sought to investigate whether it is possible to reverse the cell proliferation in resistant breast cancer cells by inhibiting certain DUSPs. We previously developed a series of Boolean models of DUSP regulation [3], and showed that they were in qualitative agreement with gene expression data of DUSPs in breast cancer cell lines treated for 24 h with Herceptin. However, Boolean models cannot distinguish between interaction intensities or protein levels. For this reason, herein we present kinetic models of DUSP regulation and investigate how the modulation of active DUSP levels affects cellular proliferation in the presence or absence of Herceptin.

We reported previously [20] that DUSP8 and DUSP16 are overexpressed in Herceptin-resistant cells but, as reported in the literature [21], it is unknown by which of the MAP kinases they are induced. Using Boolean models, we were able to predict that ERK1/2 may be an inducer for these DUSPs, hence these DUSPs are important candidates for further evaluation by kinetic modelling [3]. 

Here, we built a series of kinetic models incorporating HER2, MAP kinases, DUSPs (Figure 1), and their regulatory interactions, as described in the Methods. We used these models to investigate whether, by inhibiting certain DUSPs, Herceptin-resistant breast cancer cells would be re-sensitized to the drug. 

### 3.2. DUSP Inhibition Can Reverse Cell Proliferation

Next we used a kinetic modelling approach to investigate whether the inhibition of DUSP16 has the potential to reduce cellular proliferation in Herceptin-resistant cells treated with Herceptin. This reduction could indicate the potential of inhibiting DUSP16 to reduce or abrogate resistance to Herceptin in the cells.

Figure 2 shows results of time-course simulations of our ODE-based models for different strengths of inhibition of DUSP16. The strength of inhibition was modulated by changing the value of *d*_5_ in the kinetic model, which defines the rate of DUSP inactivation. With no inhibition or with weak inhibition of this phosphatase (Figure 2A–C), survival continuously increased over time. This means that signals that promote cellular proliferation continued to support tumor cell growth under these conditions. However, when DUSP16 was more strongly inhibited, the survival function gradually decreased. With *d*_5_ = 50 (Figure 2D), the survival function became inflected, and with higher values of *d*_5_ it decreased, indicating a reversal of cell proliferation. This effect was due to increasing activity levels of both p38 and JNK, which oppose cell proliferation.

### 3.3. Stable Knockdown of DUSPs in HER2-Positive Breast Cancer Cells Affects Cell Numbers

In order to experimentally determine the role of DUSPs in Herceptin-resistant and -sensitive breast cancer cells, we used shRNA to generate stable knockdowns of DUSPs. Scrambled shRNA (NT), which does not target any RNA, was used as a negative control. We tested several shRNA vectors for each DUSP, and selected the ones that induced decreases of more than 70% in the expression of the *DUSP* mRNA that they targeted. Figure 3 shows the expression of *DUSP8* and *DUSP16* genes in their respective stable knockdowns with the selected shRNA vectors, in both sensitive and resistant SK-BR-3 cells. The gene expression levels were significantly decreased in the respective knockdowns.

In order to investigate the effect of DUSP8 and DUSP16 knockdown in Herceptin-sensitive and -resistant cells, we compared how Herceptin affects numbers of the DUSP silenced cells with cells in which DUSPs had not been silenced. Aliquots of both Herceptin-sensitive and -resistant cells of each category were exposed to 50 µM Herceptin for 72 h, and the changes in their numbers during this period were quantified.

When Herceptin-resistant DUSP8-silenced breast cancer cells were treated for 72 h with Herceptin, the cell count continued to increase compared to cells infected with a control, non-targeting shRNA (NT). (Figure 4A). This observation suggests that silencing of DUSP8 is not able to reverse resistance. However, when Herceptin-resistant DUSP16-silenced breast cancer cells were treated for 72 h with Herceptin, a significant decrease in cell count was measured compared to NT (Figure 4B). This observation is in agreement with the outcome of the kinetic model, where DUSP16 was able to reduce tumor cell proliferation when no inducer was applied (Figure 2).

### 3.4. Models of Potential Regulation of DUSPs by MAP Kinases

The model presented in Figure 1 did not explain the observed increase in cell count in DUSP8-silenced breast cancer cells. Therefore, we tested whether the model could be modified to reflect our experimental findings by adding new inducers for DUSP8. When DUSP8 regulation by ERK12 or JNK12 was added to the model, the survival function did not decrease, even when high inhibition strength was applied to DUSP8 (Figure 5 and Figure 6). This result agrees with experimental measurements in DUSP8-silenced cells (Figure 4A) and suggests that DUSP8 may be regulated by one of these MAP kinases.

In order to discriminate between the regulation of DUSP8 by ERK12 or JNK12, we constructed two additional models in which DUSP8 is induced either by ERK12 (Figure 5) or by JNK12 (Figure 6). When ERK12 induced DUSP8, the level of active phosphatase remained high over time even for high values of *d*_5_, representing strong inhibition, for example by a potential drug (Figure 5). In contrast, when DUSP8 was regulated by JNK12, its active level decreased much more rapidly for low values of *d*_5_.

## 4. Discussion and Conclusions

Computational modelling has become widely used for biological research in recent years. Through the development of computational models, hypotheses can be tested, data can be interpreted, and predictions can be generated that assist experimental biology. There is a range of different mathematical techniques that can be used for modelling, and the choice of technique is not usually determined by the biological system under study, but by the question to be addressed. Here, our aim was to investigate whether the inhibition of certain DUSPs could be a suitable approach to re-sensitizing Herceptin-resistant breast cancer cells to the drug.

The interactions between DUSPs and cellular signaling pathways are highly specific to each DUSP, since they can be activated by different inducers, and at the same time they regulate different substrates [21]. Boolean modelling has the advantage of allowing easy testing of large numbers of network topologies, showing which of these configurations have the potential to achieve a desirable qualitative outcome. Indeed, our initial modelling studies used this approach and succeeded in reducing the number of candidate DUSPs for inhibition [20]. Nevertheless, Boolean modelling only leads to qualitative outcomes both in terms of the amounts of compounds and time scales of processes involved in the biological system. Therefore, we developed ODE-based models for the most promising targets and quantitatively investigated the potential effects of their potential inhibition.

As a first step, we showed that the qualitative behavior predicted by the Boolean model was confirmed for DUSP16, but that the rate of decrease in cell numbers was strongly dependent on the intensity of the applied inhibitor (Figure 2). This result was not unexpected, since the Boolean and kinetic models are based on the same network topology, but it highlights the added value of a quantitative method. In comparison with experimental analyses, we observed that Herceptin-resistant DUSP16-silenced breast cancer cells became more responsive to the drug when treated for 72 h with Herceptin, which agreed with the model prediction (Figure 4).

However, applying the same approach to DUSP8, cell numbers were not decreased experimentally following Herceptin treatment of Herceptin-resistant DUSP8-silenced breast cancer cells. This indicated that the model used for DUSP16 regulation was not sufficiently accurate to understand DUSP8 regulation. Since the inducers of DUSP8 are unknown, we hypothesized that its potential substrates, ERK1/2 or JNK1/2, could serve as inducers. Following simulations using two new models, we observed that a stronger reduction in the amount of DUSP8 could be achieved in the scenario where DUSP8 was induced by ERK1/2 rather than by JNK1/2. Hence, these results enabled us not only to interpret the experimental observations in Herceptin-resistant DUSP8-silenced cells, but also to strengthen the hypothesis that DUSP8 may be regulated by ERK1/2.

The models presented here link increased cell survival with increased cell proliferation. However, the experimentally measured changes in cell numbers could also arise by other mechanisms, such as changes in cell survival. Further studies are required to clarify this point.

To our knowledge, no kinetic modelling of DUSP dynamics in breast cancer has previously been reported. Overall, this study provides a useful illustration of the benefits of a systems biology approach to understanding signaling mechanisms involved in cancer development. We showed that the combination of several modelling techniques adds value, as these techniques have different benefits and provide complementary levels of interpretation. Moreover, the interplay between hypothesis generation, model development, and experimental analysis highlights the benefits of this interdisciplinary approach to biological research.

## Figures and Tables

**Figure 1 genes-10-00568-f001:**
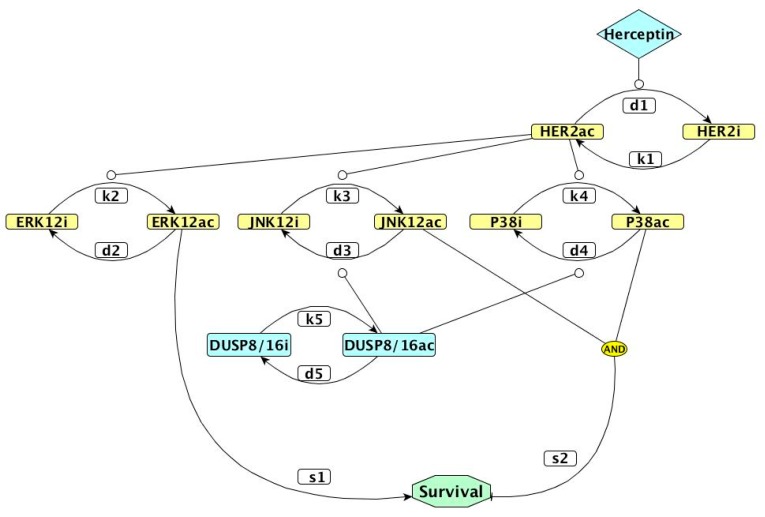
Network representation of the kinetic model used in our study. Colored rectangles represent proteins, with *ac* corresponding to the active form and *i* to the inactive form. The white rectangles represent kinetic parameters. The diamond represents Herceptin uptake and octagonal shape is the survival outcome. Pointed arrows represent activation, blunt arrows represent inhibition, and circle-shaped arrows represent the stimulation of activating or inhibitory transitions. DUSP: dual-specificity phosphatase; ERK12: extracellularly regulated kinases 1 and 2; JNK12: c-Jun N-terminal kinases 1 and 2.

**Figure 2 genes-10-00568-f002:**
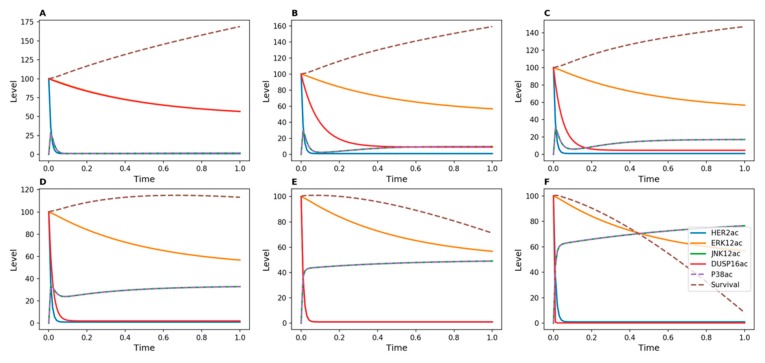
Kinetic simulations showing effects of Herceptin on Herceptin-resistant cells when DUSP16 is inhibited. No inducer of DUSP16 was added in the model. DUSP16 inhibition was achieved by modulating the value of the *d*_5_ parameter in the kinetic model, which represents the transition from active to inactive DUSP: (**A**) *d*_5_ = 1; (**B**) *d*_5_ = 10; (**C**) *d*_5_ = 20; (**D**) *d*_5_ = 50; (**E**) *d*_5_ = 100; (**F**) *d*_5_ = 500.

**Figure 3 genes-10-00568-f003:**
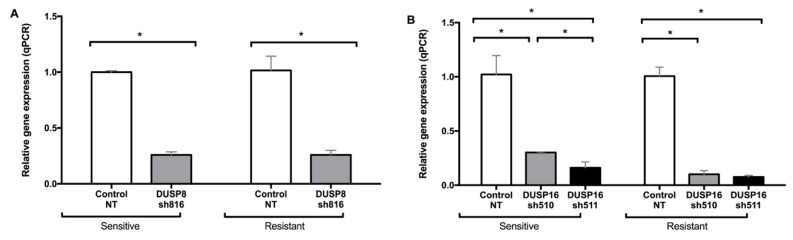
Dual-specificity phosphatase (DUSP) stable knockdowns in Herceptin-sensitive and Herceptin-resistant SK-BR-3 cells using selected shRNA (small hairpin RNA) vectors for *DUSP8* (**A**) and *DUSP16* (**B**). Data represent means ± SE (standard error) of two independent experiments. Asterisks indicate statistically significant differences of *p* < 0.05. NT: non-targeting shRNA. Cells had previously been grown in culture for six months in the absence or presence of 50 µM Herceptin (marked as Sensitive or Resistant, respectively).

**Figure 4 genes-10-00568-f004:**
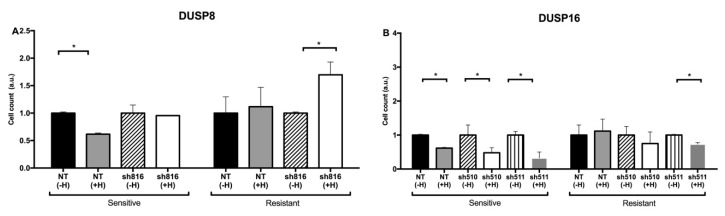
Relative numbers of Herceptin-sensitive or -resistant SK-BR-3 cells in the presence of Herceptin following knockdown of *DUSP8* (**A**) or *DUSP16* (**B**). As detailed in the Methods, the increases in cell numbers during a period of 72 h following seeding in the presence or absence of Herceptin were determined. Shown are the cell numbers after 72 h, divided by the starting cell numbers and—within each pair of sensitive vs. resistant cells—normalized to the value obtained for the sensitive cells. Bars represent mean ± SE cell numbers in two independent experiments. Asterisks indicate statistically significant differences of *p* < 0.05.

**Figure 5 genes-10-00568-f005:**
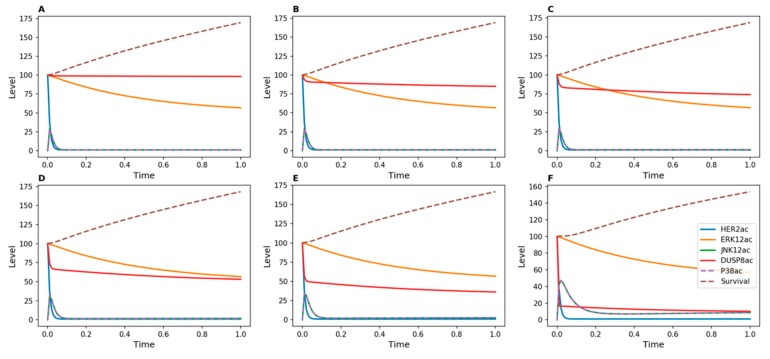
Kinetic simulations showing effects of Herceptin in Herceptin-resistant cells when DUSP8 was inhibited, with DUSP8 inducible by ERK12. DUSP8 inhibition was achieved by modulating the value of the *d*_5_ parameter in the kinetic model, which represents the transition from active to inactive DUSP. (**A**) *d*_5_ = 1; (**B**) *d*_5_ = 10; (**C**) *d*_5_ = 20; (**D**) *d*_5_ = 50; (**E**) *d*_5_ = 100; (**F**) *d*_5_ = 500.

**Figure 6 genes-10-00568-f006:**
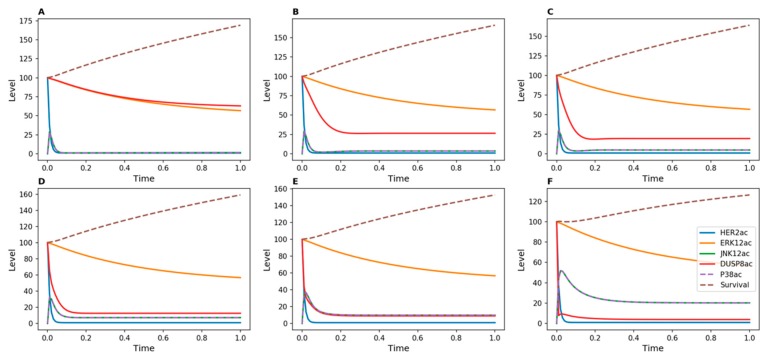
Kinetic simulations showing the effects of Herceptin in Herceptin-resistant cells when DUSP8 was inhibited, with DUSP8 inducible by JNK12. DUSP8 inhibition was achieved by modulating the value of the *d*_5_ parameter in the kinetic model, which represents the transition from active to inactive DUSP. (**A**) *d*_5_ = 1; (**B**) *d*_5_ = 10; (**C**) *d*_5_ = 20; (**D**) *d*_5_ = 50; (**E**) *d*_5_ = 100; (**F**) *d*_5_ = 500.

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
