# Peer review of "Kinetic Modeling of DUSP Regulation in Herceptin-Resistant HER2-Positive Breast Cancer"

_genes, 2019, doi:10.3390/genes10080568_

Round 1

Reviewer 1 Report

Line 130, "0.5 106 cells for each repeat" is hard to understand.

Author Response

Thank you for reviewing our manuscript. We have corrected line 130 (0.5 x 10^6) and highlighted it in red.

Reviewer 2 Report

The authors have presented a well-written and concise manuscript about the kinetic modeling of DUSP regulation in herceptin-resistant HER2-positive breast cancer. 

They investigated whether inhibiting certain DUSPs re-sensitizes herceptin-resistant breast cancer cells to the drug, and observed that herceptin-resistant DUSP16-silenced breast cancer cells became more responsive to the drug when treated for 72 hours with Herceptin, showing a decrease in resistance.

I have no major concerns about the manuscript, but will like the authors to edit the manuscript for typos such as seen in line 288: "...showing which of these configurations have to potential to achieve a desirable qualitative outcome." Also, the discussion should include the comparison of the results of this current study with those of similar studies. What are the limitations of this study?

Author Response

Thank you for reviewing our paper. We have corrected the typo in line 286 and added a sentence indicating that to our knowledge no kinetic modelling of DUSP dynamics in breast cancer has been reported before. The limitations of the study are presented in lines 308-310. Changes are highlighted in red.